# Linking Metabolic Reprogramming, Plasticity and Tumor Progression

**DOI:** 10.3390/cancers13040762

**Published:** 2021-02-12

**Authors:** Oleg Shuvalov, Alexandra Daks, Olga Fedorova, Alexey Petukhov, Nickolai Barlev

**Affiliations:** 1Institute of Cytology RAS, 194064 St-Petersburg, Russia; oleg8988@mail.ru (O.S.); alexandra.daks@gmail.com (A.D.); fedorovaolgand@gmail.com (O.F.); alexeysakhalin@gmail.com (A.P.); 2Almazov National Medical Research Center, 197341 St-Petersburg, Russia; 3MIPT, 141701 Dolgoprudny, Moscow Region, Russia; 4Orekhovich IBMC, 119435 Moscow, Russia

**Keywords:** metabolic reprograming, cancer metabolism, aerobic glycolysis, lipid metabolism, one-carbon metabolism, cancer therapy

## Abstract

**Simple Summary:**

In the present review, we discuss the role of metabolic reprogramming which occurs in malignant cells. The process of metabolic reprogramming is also known as one of the “hallmarks of cancer”. Due to several reasons, including the origin of cancer, tumor microenvironment, and the tumor progression stage, metabolic reprogramming can be heterogeneous and dynamic. In this review, we provide evidence that the usage of metabolic drugs is a promising approach to treat cancer. However, because these drugs can damage not only malignant cells but also normal rapidly dividing cells, it is important to understand the exact metabolic changes which are elicited by particular drivers in concrete tissue and are specific for each stage of cancer development, including metastases. Finally, the review highlights new promising targets for the development of new metabolic drugs.

**Abstract:**

The specific molecular features of cancer cells that distinguish them from the normal ones are denoted as “hallmarks of cancer”. One of the critical hallmarks of cancer is an altered metabolism which provides tumor cells with energy and structural resources necessary for rapid proliferation. The key feature of a cancer-reprogrammed metabolism is its plasticity, allowing cancer cells to better adapt to various conditions and to oppose different therapies. Furthermore, the alterations of metabolic pathways in malignant cells are heterogeneous and are defined by several factors including the tissue of origin, driving mutations, and microenvironment. In the present review, we discuss the key features of metabolic reprogramming and plasticity associated with different stages of tumor, from primary tumors to metastases. We also provide evidence of the successful usage of metabolic drugs in anticancer therapy. Finally, we highlight new promising targets for the development of new metabolic drugs.

## 1. Introduction

Specific molecular features of cancer cells that distinguish them from the normal ones are called “hallmarks of cancer” [1]. In general, eleven “hallmarks of cancer” can be outlined, including sustained proliferative signaling, resistance to cell death, evasion from growth suppressors, evading the immune system-mediated destruction, the ability to invade and metastasize, induce angiogenesis, genomic instability, replicative immortality, tumor-promoting inflammation, and deregulated cellular energetics [1]. Importantly, all therapies that have been developed to date are directed toward the “hallmarks of cancer”.

One of the critical hallmarks of cancer is altered metabolism, which provides tumor cells with the energy and structural resources necessary for rapid proliferation. The metabolic hallmark is multifaceted and defines the spectrum of metabolic alterations inherent to neoplastic cells. Such metabolic alterations are collectively referred as “metabolic reprogramming” [2]. It mainly includes increased aerobic glycolysis (Warburg effect) which cooperates with oxidative phosphorylation (OXPHOS) to establish a “hybrid” metabolic state, deregulated tricarboxylic acid (TCA) cycle, lipid and glutamine metabolism, and increased one-carbon metabolism. It seems that the main feature of cancer-related metabolism is not just an enhanced intensity of a particular metabolic pathway, but rather the overall plasticity of the regulation of metabolic pathways [3].

During cancer development, malignant cells pass through several stages: cancer initiation, tumor growth, intravasation into blood or lymph vessels, and dissemination across the body, extravasation back to the appropriate niches and their colonization, etc. This complicated cycle is full of different challenges. Tumor formation is physiologically and genetically associated with and is limited by the tissue of origin and proceeds in a specific microenvironment [4]. The speed of tumor growth is controlled by different availabilities of nutrients and oxygen, which in turn depends on vascularization of the particular tumor. Additionally, as tumors grow, they become more heterogenous due to the Warburg effect and related acidosis. Furthermore, they have to withstand the aggressiveness of immune cells that constant surveil to combat tumor cells [5]. In response to different regulatory cues, the primary tumors become metastatic. Spreading metastasis involves the process called epithelial-to-mesenchymal transition (EMT), which is associated with gross morphological changes to lose epithelial features to become pseudomesenchymal and invasive [6]. The EMT cells undergo oxidative shock upon intravasation of blood vessels, followed by constant attacks from the immune cells in the blood stream. To survive, they switch their glycolytic metabolisms to the hybrid state, which allows them to adapt and, after extravasation, establish metastases [3].

Finally, when the tumor cells reach an appropriate niche, they extravasate and undergo mesenchymal-to-epithelial transition to colonize a new milieu (Figure 1). To fit this complex and changing environment, cancer cells have evolved a high degree of plasticity in regulation of their proliferation, mobility and invasion, antioxidative responses, resistance to apoptosis, and evasion from the immune response. The molecular basis for such plasticity is provided by reversible metabolic changes in cancer cells that supply them with energy and structural resources (“building blocks” for protein synthesis, glutathione and Nicotinamide adenine dinucleotide (NADH) for redox homeostasis, S-adenosyl methionine (SAM) for epigenetic regulation of gene expression, etc.). Therefore, the plasticity of metabolic regulation of transformed cells is considered as an important “hallmark of cancer” (Figure 1).

At present, deregulated metabolic properties are widely recognized as therapeutic liabilities for anticancer therapy. The therapeutic importance of this approach is signified by more than 70 years of successful implementation of antimetabolite therapy targeted at one-carbon metabolism [7]. The reported data regarding the onco-associated metabolic features include the “hybrid” Warburg/OXPHOS energy phenotype, the deregulated TCA cycle, and altered glutamine, lipid, and one-carbon metabolism. Deciphering the regulatory mechanisms of these reactions has identified the corresponding enzymes which are considered as promising targets for pharmacological inhibition.

## 2. Specific Metabolic Alterations in Malignant Cells

Generally, the key metabolic features of cancer cells include the elevated uptake of glucose and glutamine, in parallel with increased one-carbon metabolism. The ability to utilize glycolysis, OXPHOS, and β-oxidation of fatty acids satisfies the energy demand of cancer cells in changing environmental conditions [8].

In mammalian cells, energy production is based on glycolysis and oxidative phosphorylation—two processes that are coupled and cooperate to supply energy. Glycolysis is a more evolutionary ancient process compared to OXPHOS and took place in ancient bacteria under anaerobic conditions. During this process, glucose is converted into pyruvate concomitantly, producing two molecules of adenosine triphosphate (ATP) per one molecule of glucose. In aerobic conditions, pyruvate enters the TCA cycle, thus coupling glycolysis with OXPHOS that produces 36 molecules of ATP. In anaerobic conditions, pyruvate is converted to lactate which is then removed from the cell. Normally, the energy demand of cells is supplied up to 70% by OXPHOS [9]. However, cancer cells preferentially use glycolysis even in aerobic conditions. This phenomenon was called the “Warburg effect” and has been known since 1920s. Otto Warburg hypothesized that malignant cells display high rates of aerobic glycolysis due to the deficit of mitochondria [10]. 

Now it is well-known that defects in mitochondria that disrupt the OXPHOS process are not typical for different neoplasms. In most cases, respiration remains intact in malignant cells [3,11,12], Importantly, cancer cells can switch from glycolysis to OXPHOS and back under different environmental conditions—for instance, acidosis or radiotherapy [13,14,15,16,17]. This dual metabolic phenotype (aerobic glycolysis and OXPHOS) of malignant cells has led to the hypothesis of a “hybrid” metabolic state. This hybrid phenotype increases the metabolic capacity of cancer cells and makes them more resistant to aggressive and changing conditions. One illustration of this metabolic plasticity is the switch from glycolysis to the hybrid state during EMT. When glycolytic cancer cells intravasate blood vessels [4], they undergo oxidative stress. 

Moreover, aerobic glycolysis is such a frequently detected phenomenon in cancer cells that it is used in diagnostics of tumors and metastases by detecting labeled fluorodeoxyglucose ({18F}-FDG) by positron-emission tomography [18]. The high level of aerobic glycolysis in cancer cells is the result of the balance between the activities of oncogenes, tumor suppressors, and tumor microenvironment. Cancer cells benefit from a high glycolytic rate because it efficiently provides “building blocks” for the needs of anabolism, contributes to redox homeostasis, and makes the microenvironment more “friendly” to their inhabitation (for more information, see the comprehensive review of [19]). 

Glycolysis is pivotal to the anabolic requirements of rapidly proliferating malignant cells because it diverts the glucose flux into pentose-phosphate pathway (PPP) to generate serine, glycine, and one-carbon equivalents to produce purine and pyrimidine nucleotides as well as NADPH to circumvent the deleterious effect of reactive oxygen species (ROS) [20]. The key glycolytic enzyme in this respect is the pyruvate kinase isozyme PKM2. *PKM1* and *PKM2* are two spliced variants of the *PKM* gene. Pyruvate kinase converts phosphoenolpyruvate to pyruvate in the final step of glycolysis. PKM2 is expressed in embryonic tissues and proliferating cells and is then replaced by other tissue-specific pyruvate kinases. PKM2 is widely expressed in malignant cells providing them with a growth advantage without the production of ROS [21]. Its dimer possesses low catalytic activity, resulting in the rerouting of the glycolytic flux to PPP, thus providing tumor cells with an additional source of anabolism [22].

On the contrary, downregulation of glycolysis [23] or induction of mitochondria biogenesis [24] lead to the restoration of OXPHOS and subsequent death of malignant cells. Despite this common view, there are experimental data suggesting that high levels of OXPHOS are important to many cancer cells models and tumors [25,26].

Similar to attenuation of glycolysis, inhibition of respiration may also represent a therapeutic strategy to eliminate the growth of malignant cells [27]. Emphasizing the importance of respiration for cancer cells, Farge and colleagues [16] have shown that citarabine-resistant acute myeloid leukemia cells were not enriched for leukemic stem cells but relied on oxidative metabolism. OXPHOS is also required for maintaining stemness in hepatocellular carcinoma (Liu, 2020). Targeting OXPHOS restores the sensitivity of vemurafenib (v-Raf murine sarcoma viral oncogene homolog B (BRAF) inhibitor) - and gefitinib (Epidermal growth factor receptor (EGFR) inhibitor)-resistant cancer cells to the corresponding drugs [17].

It is now widely accepted that glycolysis cooperates with OXPHOS to provide metabolic plasticity to cancer cells [11]. Importantly, the downregulation of glycolysis may result in upregulation of OXPHOS in order to obtain the ATP necessary for survival of cancer cells [28,29], suggesting that these two processes are inter-related.

It has been hypothesized that the metabolic switch is a stepwise process. Several “waves” of transcriptional activation of genes responsible for critical metabolic enzymes occur during carcinogenesis before the cancer cells acquire the glycolytic phenotype. The high proliferation rate creates aglycemic conditions, resulting in stimulation of glutaminolysis, which in turn, restores the OXPHOS process [30]. Thus, metabolic plasticity allows cancer cells to utilize a dynamic cooperation between glycolysis and OXPHOS to adapt to environmental changes [19].

Dissecting the regulatory mechanisms of glycolysis/OXPHOS plasticity is very important for developing new anticancer therapeutics. In this respect, a number of studies have shown that c-Myc plays an important role in regulation of energy balance by affecting both glycolysis and OXPHOS (for review, see [31]). In addition to augmenting the vast majority of glycolytic gene products, c-Myc also elevates the mitochondrial mass and oxygen consumption [32]. On the contrary, downregulation of c-Myc results in mitochondria defects [33]. These results define c-Myc as one of the master-regulators of the glycolysis/OXPHOS switch.

Furthermore, the glycolytic enzyme, pyruvate kinase PKM2, coordinates glycolysis and OXPHOS by interacting with mitofusin 2 (MFN2), a key regulator of mitochondrial fusion and the OXPHOS-dependent attenuation of glycolysis [34].

By using the mathematical modeling approach combined with metabolomics and gene expression data, Jia with colleagues [15] has demonstrated a direct association between the activities of AMP-activated protein kinase (AMPK) and Hypoxia-inducible factor 1 (HIF-1) (master-regulators of OXPHOS and glycolysis, respectively) and glycolysis/respiration. They have shown that triple negative breast cancer (TNBC) cells can maintain a hybrid metabolic phenotype and that targeting both glycolysis and OXPHOS is necessary to eliminate their metabolic plasticity.

Glutamine is the most abundant amino acid in the human body [35]. Rapidly proliferating cells, the malignant ones in particular, use glutamine to both supply their energy demands (through TCA cycle) and as a source of carbon and nitrogen for building the biomass [36]. During the process called “gluthaminolysis”, gluthamine is converted to glutamate which is then reduced to α-ketoglutarate. α-Ketoglutarate enters the TCA cycle and can be further oxidized supplying the electron-transfer chain or undergoes reducing carboxylation to produce citrate, which in turn supports the synthesis of lipids through acetyl-coenzym A (acetyl-CoA) (so called anaplerosis) (for a comprehensive review, see [37]). Thus, glutamine mediates metabolic reprogramming by both supporting mitochondrial oxidative phosphorylation and supplying metabolic intermediates for the TCA cycle. Notably, glutamine can be the major fuel for the mitochondrial respiration in pancreatic cancer cells and mammalian fibroblasts transformed with Ras or Akt [38,39]. Glutamine can also be used as a substrate for enzymes involved in the de novo synthesis of purine and pyrimidine nucleotides [40]. Furthermore, the impact of glutamine on the nucleotide metabolism and DNA repair determines resistance of cancer cells to radiotherapy [41]. Enhanced glutamine metabolism is associated with hypoxia-mediated chemoresistance [42] and resistance of pancreatic cancer cells to gemcitabine. The latter, at least partially, is based on glutamine-mediated glutathione production [43].

It was shown that lung squamous cell carcinoma adapts to mTOR inhibition and suppression of glycolysis through the GSK3α/β signaling pathway, which in turn enhances glutaminolysis (Momcilovic 2018). This adaptive glutamine catabolism sensitizes cancer cells to glutaminase inhibition. Collectively, these facts suggest that glutamine is one of the key mediators of the adaptive metabolic changes that takes place in cancer cells.

The importance of glutamine for cancer cell metabolism and survival is proved by successful clinical application of CBT-839, the inhibitor of glutaminase (GLS), which “enters” glutamine to open cancer metabolism [44,45]. 

An additional metabolic “hallmark of cancer” is lipidomic remodeling, which includes alterations in the fatty acids transport, their de novo synthesis, storage as lipid droplets, and β-oxidation to generate ATP (for review, see [46]). 

In normal cells, de novo synthesis of fatty acids is restricted to liver and adipose tissues. However, in malignant cells, this anabolic pathway is reactivated [46]. Specifically, acetyl-CoA carboxylase (ACC) mediates condensation of acetyl-CoA with malonyl-CoA, resulting in the synthesis of palmitate (16:0). Then, fatty acid synthase (FASN) builds up the chain by repeatedly adding the malonyl-CoA monomers. Cancer cells utilize lipid synthesis to supply their demand in structural and signaling components. ACC and FASN are frequently overexpressed in various malignancies [47,48]. On the contrary, their downregulation leads to the decrease in cancer cell proliferation and induction of apoptosis. The biosynthesis of lipids is a controlled molecular program regulated by the sterol regulatory element binding proteins (SREBPs), which respond to upstream signaling cues (i.e., PI3K/AKT/mTORC1 pathway) as well as the levels of cellular nutrients to transcribe genes that code for enzymes responsible for cholesterol and fatty acid uptake and biosynthesis [36]. SREBPs are overexpressed in various malignancies and their downregulation leads to the inhibition of tumor growth [49,50,51].

In addition to glycolysis and OXPHOS, there is another source of energy engaged by cancer cells—i.e., β-oxidation of fatty acids (FAO). Fatty acids can be the source of energy in response to fluctuating microenvironmental conditions such as acidosis, which inhibits glycolysis [52]. A critical enzyme for this pathway is Carnitine Palmitoyltransferase (CPT-1), a rate-limiting enzyme which controls oxidation of fatty acids. It links the acyl-group of a long-chain fatty acyl-CoA from coenzyme A with carnitine, allowing fatty acids to be transported from the cytosol into the intermembrane space of mitochondria. The overexpression of CPT-1A is frequently associated with tumor progression in ovarian, lung, breast, gastric, and prostate cancers as well as in blood malignancies [53,54].

Moreover, metabolic cross-talk occurs between cancer cells and adipocytes, resulting in a more aggressive phenotype of malignant cells [55]. Cancer-associated adipocytes represent a prominent source of external lipids for cancer cells. The latter can retrieve fatty acids from adipocytes through releasing soluble signaling factors (for review, see [56]).

In this respect, it is interesting to note that the inhibition of lipid biosynthesis in colon cancer cell models dampens both respiration and FAO. It has been suggested that levels of glucose metabolism are dictated by requirements in lipid production in cancer cells, and fatty acids derived from the de novo lipid biosynthesis are used to fuel tumor growth [49]. 

Furthermore, lipid metabolic reprogramming contributes to the chemotherapy resistance by a number of mechanisms (for review, see [56]). It diminishes the oxidative stress elicited by anticancer therapeutics [57,58], interferes with drug-induced cell death [59], and favors the production of drug-resistant cancer stem cells [60]. Thus, lipid metabolic reprogramming contributes to the metabolic flexibility that is adapted by cancer cells to fulfill the energy requirement required for proliferation and survival.

Finally, another critical metabolic alteration in malignant cells is an increased one-carbon metabolism. It perceives and controls the nutrients status of cells through cycling of one-carbon groups and redistributing them between different acceptor molecules. The donors of one-carbon groups are either glycine or serine amino acids. One-carbon metabolism controls synthesis of nucleotides, certain amino acids, S-adenosylmethionine (SAM), glutathione, etc., which are the acceptors of one-carbon groups [7,61]. All of these compounds are pivotal for rapidly proliferating malignant cells.

## 3. Molecular Drivers of Metabolic Changes in Cancer Cells

Importantly, major oncogenes not only rewire the gene expression programs of neoplastic cells (Figure 2), but are also responsible for metabolic reprogramming. For example, mutations in the Kirsten Rat Sarcoma virus (KRAS) gene is one of the most common drivers in lung, colon, and pancreatic cancers [62]. When mutated, KRAS becomes uncontrollably active and promotes proliferation of cells via the RAS/MAPK pathway. The KRAS-driven metabolic rewiring is mediated by the upregulation of several rate-limiting enzymes involved in amino acid, fatty acid, or nucleotide biosynthesis [63]. Unfortunately, due to its structure, KRAS still remains an undruggable target [64].

In non-small cell lung cancer (NSCLC) [65] as well as in colorectal cancer [66], glucose transporter (GLUT1) is overexpressed in KRAS-mutated lung carcinomas in comparison with cells bearing wild-type KRAS. GLUT1 is a key glucose transporter required for enhanced glucose uptake by cancer cells. The increased expression of GLUT-1 promotes cell viability and is associated with poor outcome of cancer therapy [67]. Thus, GLUT-1 is an important player in cancer cell proliferation, migration, and invasion [68].

In addition, an increase in the copy number of mutant KRAS(G12D) increases glucose consumption, glycolytic gene expression, and lactate secretion. The overexpression of mutant KRAS induces glycolytic switch coupled to increased channeling of glucose-derived metabolites into the TCA cycle and glutathione biosynthesis, resulting in enhanced glutathione-mediated detoxification [69].

On a related note, the KRAS-dependent PI3K-Akt-mTOR pathway affects the uptake and utilization of multiple nutrients, including glucose, glutamine, nucleotides, and lipids [70]. Being a kinase, Akt controls glucose uptake, activity of glycolytic enzymes as well as enzymes of lipid metabolism at both post-translational and transcriptional levels (Table 1). 

As part of the proliferation cue, Akt induces expression of c-Myc and Hif1a—two other key oncogenes, which trans-activate genes coding for metabolic enzymes. As mentioned above, c-Myc is a well-known master-regulator of metabolic networks. It induces changes in glucose, glutamine, one-carbon, and lipid metabolism [31,75] (Table 1). 

It seems obvious that inhibiting c-myc in cancer cells would dramatically benefit the patients. However, there are no c-myc-specific drugs available on the market yet and the perspective peptide mimic of c-myc is currently only in the initial phase of clinical trials [93]. Thus, the only current viable pharmacological options on hand are to dissipate the relevant oncogenic pathways and to target the metabolic consequences of deregulated cancer-driving genes.

Importantly, another oncogenic transcription factor, HIF1a, which is stabilized in response to oxygen deprivation, also dramatically affects glycolysis, TCA cycle, PPP, and glutamine metabolism [94,95,96] (Table 1). Hypoxia usually occurs in solid tumors as a consequence of incomplete and disordered vasculature, which is developed to supply oxygen to the rapidly growing tumor. Paradoxically, a number of studies have shown that approximately one half of tumors without hypoxia also display HIF1a stabilization. Apparently, stabilization of HIF1a is critical for cancer cells because it induces metabolic adaptations favoring tumor development and progression. 

FoxO transcription factors (TFs) represent an important component of the insulin signaling pathway and display oncosuppressor features. FoxO factors are inhibited by insulin through the activation of PI3K-Akt signaling. Akt-mediated phosphorylation inactivates FoxO proteins, thereby inhibiting apoptotic and antiproliferative transcriptional programs, and thus linking diabetes and cancer [97]. It is thought that FoxO attenuates the Warburg effect and glucose uptake at least partly by antagonizing c-Myc by several mechanisms: by directly binding the promotor regions of c-Myc responsive genes, by augmenting the transcriptional repressors of the MAD/MXD family, and by facilitating the expression of microRNAs that target c-Myc [98,99,100]. Some of FoxO-mediated effects on metabolism are summarized in Table 1.

## 4. Metabolic Features of Primary Malignancies

It is important to note that the initial steps of tumorigenesis occur in the metabolic context of the tissue of origin [4]. This means that both the cancer-driving mutations in tumor-initiating cells and the tissue of origin set specific metabolic features for the tumor. This is one of the factors that drives metabolic heterogeneity of cancer. 

Upon the growth of tumor mass, neoplastic cells acquire metabolic alterations which provide them with abilities for rapid proliferation, resistance to apoptosis, radio- and chemotherapy, etc. In other words, malignant cells fine-tune their metabolisms to support the increased energy demand in order to adapt to rapid growth and survival in their environment [101]. 

It is known that the aberrant expression or the appearance of mutations in a number of oncogenes or tumor suppressors can also instigate metabolic reprogramming (e.g., c-Myc, KRAS, PI3K, TP53, etc.). It is tempting to speculate that cancer-initiating mutations in these driver genes are also able to induce metabolic alterations that further promote the tumor initiation [4].

To illustrate this hypothesis, it is worth mentioning isocitrate dehydrogenase (IDH), an enzyme of the Krebs cycle, which has two isoforms—IDH1 and IDH2. Mutations in IDH1 and IDH2 result in altered enzymatic activity that leads to the production of 2-hydroxyglutarate, an oncometabolite which is absent in normal (nonmutated IDHs) cells. 2-Hydroxyglutarate was shown to inhibit histone demethylases, causing the hypermethylation phenotype [102]. In turn, this phenotype is associated with impaired hematopoietic differentiation and increased expression of stem/progenitor cell markers, promoting a proleukemogenic effect [103]. 

Moreover, germline mutations in the *SDHx* gene, which codes for another enzyme of the TCA cycle, can drive rather rare malignancies: paraganglioma, pheochromocytoma, gastrointestinal stromal tumors, renal cell carcinomas, and other endocrine-related tumors [104]. Further, mutations in another TCA cycle enzyme, fumarate hydratase (FH), cause hereditary leiomyomatosis and renal cell cancer [105]. Mutations in SDHx and FH result in the accumulation of succinate and fumarate, respectively [106,107]; mechanistically, 2-hydroxyglutarate and the increased accumulation of fumarate and succinate disrupt the functions of enzymes involved in the maintenance of epigenetic homeostasis. However, despite known germline mutations in *IDHx, SDHx*, and *FH*, tumors associated with these mutations are rare. This means that the tissue of origin can be the main restriction factor to oncogenesis, limiting the malignant penetration of particular drivers.

Another interesting question is how metabolism itself is linked to primary carcinogenesis. It is well-established that the level of glucose, obesity, and type 2 diabetes mellitus are all associated with enhanced risk of cancer [108,109,110]. High insulin levels (hyperinsulinemia) affect cancer development by directly stimulating the insulin receptor (IR) and/or indirectly through increasing the level of circulating Insulin Like Growth Factor 1 (IGF-1). Importantly, cancer cells frequently overexpress insulin and IGF-1 receptors [111]. Additionally, the high glucose level itself may induce proliferation and migration of tumor cells [112] by providing nutrients [113], inducing ROS [114], and activating EGFR [115]. Finally, diabetes may induce the long-term proinflammatory condition, which in turn, can cause genetic instability that is also associated with cancer [111]. Moreover, caloric restriction is a well-known factor decreasing the cancer incidence linking metabolism and cancer initiation [116,117].

Another important aspect of cancer metabolism is that hypoxia, the Warburg effect, and acidosis are linked together and favor the initiation and progression of neoplastic growth. 

For instance, there is clear evidence on the influence of hypoxia on cancer initiation [118]. Hypoxia activates several antiapoptotic proliferative signaling pathways including the ones of HIFa, PI3K-Akt-mTOR [119,120], Ras-MEK-ERK [121], and NF-kB [122]. In some cases, Hif1a cooperates with c-Myc to induce the expression of glycolytic genes [123,124,125]. Taking these facts together, it is apparent that hypoxia is a very strong inducer of glycolysis. Importantly, increased glycolysis leads to acidosis. In the early stage of growth, tumors become stratified, whereby the most aggressive cells develop within the acidic part of the tumor [126]. Acid adaptation gives rise to a more invasive phenotype, inducing metastasis [127].

Following the initiation step, the growing tumors require “building blocks” and energy production for intense proliferation, which results in enhanced anabolic metabolism. Dividing cells primarily depend on high levels of proteins, lipids, and nucleotides to create membranes and double their DNA. Nucleotide biosynthesis is an attractive target for pharmacological interference and hence represents an Achilles tendon of tumors [7,117]. In addition, one-carbon metabolism, which fuels nucleotide biosynthesis, is also an attractive anticancer drug target that has been successfully explored for approximately fifty years [128].

## 5. Epithelial–Mesenchymal Transition (EMT) and Associated Metabolic Features

Metastasis, but not the primary tumors, is the main reason for the deaths of cancer patients. Metastases invade tissues and organs interfering with their physiological functions. The initiation of metastasis is closely associated with epithelial–mesenchymal transition (EMT)—the process of losing the epithelial phenotype and molecular features by cells in order to acquire mesenchymal characteristics. EMT normally occurs during development, wound-healing, and tissue fibrosis [129]. However, cancer cells undergo only partial EMT to acquire the ability to disseminate throughout the body and establish secondary malignancies in the appropriate niche. To do this, primary cancer cells require degradation of extracellular matrix, intravasation into blood or lymph vessels, migration to new niches, escape from the immune system, and finally extravasation back from vessels. To colonize suitable niche, cancer cells undergo a reverse process to EMT, Mesenchymal–Epithelial Transition (MET) [4].

The current view on the EMT/MET suggests that this process is reversible—i.e., has plasticity. Cancer cells seldom display fully developed EMT or MET properties but rather demonstrate mixed phenotypes sharing both epithelial and mesenchymal features with one of them prevailing at the appropriate moment. For example, several studies have shown that metastatic colonization requires inhibition of EMT in extravasated cells and their conversion to the state of MET [130,131].

A number of studies have demonstrated that EMT is induced by TGFbeta, PI3K-AKT-mTOR, EGFR-RAS-MAPKs, and HIF1a [132,133,134]. Moreover, several master-regulators of EMT are broadly recognized—transcriptional factors of Zeb, Snail, and Twist families. Their expressions are associated with the loss of E-cadherin and claudins with simultaneous upregulation of mesenchymal markers—vimentin and fibronectin [129] (Figure 3).

Induction of EMT is often associated with enhanced glucose metabolism and lactate production [135]. Well-known EMT inducers (Transforming growth factor beta - TGF-beta, PI3K-AKT, KRAS, c-Myc, and HIF1a) also affect tumor metabolism. Perhaps not surprisingly, KRASmut, PI3K, and HIF1 are able to enhance glucose uptake by inducing the expressions of glucose transporters including GLUT-1 and GLUT-3 [66,136,137]. In general, a high level of GLUT-1 is associated with more aggressive and less differentiated tumors. Another important glucose transporter participating in EMT and cancer progression is GLUT-3 [115,138], whose expression is upregulated by Zeb1, TCF4/β-catenin, HMGA1, and EGFR [139,140].

A number of studies have demonstrated that master-regulators of EMT—transcriptional factors Zeb1, Snail, and Twist—are also able to regulate energy metabolism [141]. Zeb1 was shown to induce aerobic glycolysis in pancreatic cancer cell models by repressing mitochondrial-localized tumor suppressor SIRT3 [142]. Moreover, Zeb1 can directly activate GLUT-3 expression [138]. A mouse pancreatic model also demonstrates that Zeb1 depletion reduces cancer metabolic plasticity by decreasing the basal respiration and glycolytic capacity [143]. Zeb1 is also known to modulate epithelial cell adhesion by diverting glycosphingolipid metabolism [144]. Regarding noncancerous tissues, Zeb1 was demonstrated to be the central transcriptional component of adipogenic differentiation [145].

Twist was shown to enhance the glucose consumption and lactate production. It augments pyruvate kinase M (PKM2), LDHA, and glucose-6-phosphate dehydrogenase (G6PD) by activating the β1-integrin/FAK/PI3K/AKT/mTOR axis in MCF10A mammary epithelial cells [146]. Additionally, Twist is able to regulate metabolism of fatty acids [147,148], although this has not yet been shown in the context of EMT.

Similarly, Snail also affects energy metabolism. The overexpression of Snail in both pancreatic cancer cell model PANC1 and nontumorigenic human pancreatic ductal epithelial cells augmented glucose uptake, basal glycolysis, and lactate production, concomitantly reducing respiration parameters [149]. Snail-mediated increase in glucose uptake and lactate production was also shown for gastric cancer [137]. Regarding breast cancer, Kim with colleagues showed that Snail reprograms glucose metabolism by repressing phosphofructokinase PFKP which switches the glucose flux towards the pentose phosphate pathway [150]. Physiologically, this dynamic regulation of PFKP allows cancer cells to survive under metabolic stress.

EMT is strongly affected by the tumor microenvironment. For example, rapid tumor growth creates a deficit of nutrients [151], mechanistic hypertension [152], and acidosis [153,154]. In turn, acidosis drives EMT and stem-like phenotype by inducing TGF-beta [52] and stem-cell markers [150]. Using MCF7 breast cancer cells as a model, Sadeghi with colleagues (Sadeghi et al., 2020) has shown that acidosis promoted a partial EMT, conferring the cells with phenotypic characteristics of both epithelial and mesenchymal cells.

Phosphorylation of eIF2α, which is an evolutionarily conserved hallmark of the starvation response [155], triggers invasiveness in melanoma [156], breast [157], pancreatic cancer [158], and CML [159]. Taken together, these results indicate that EMT can serve as a mechanism for cancer cells to escape the worsening metabolic conditions and to colonize new niches to promote their growth.

However, it should be noted that not only limitations of nutrients can induce EMT. Apparently, high glucose levels are also able to induce EMT in the bladder cancer model via the YAP1/TAZ-GLUT1 axis [112]. Activation of this pathway results in increased migration, invasion, and metastasis formation of breast cancer cells [160]. The same pathway can trigger EMT in colorectal [161] and gastric cancer cells [162]. This means that a very complex regulatory network exists that regulates EMT upon varying environmental and metabolic conditions.

EMT can also be affected by deregulation of the TCA cycle. Mutations in its three key enzymes—fumarate hydratase (FH), succinate dehydrogenase (SDH) and isocitrate dehydrogenase (IDH)—are associated with EMT and cancer progression. The impairment of TCA cycle by mutations in these genes is associated with global perturbation of cellular epigenetics. FH mediates the conversion of fumarate to malate. Mutations in FH can elicit EMT in renal cancer cells by accumulation of fumarate which attenuates expression of the miRNA-200 cluster by inhibiting the Tet-mediated demethylation of the miR-200 regulatory region [107]. Since miR-200 targets the key EMT transcription factors, its attenuation reactivates the expression of EMT-related transcription factors and increases migration and invasion of renal cancer cells.

The loss of function mutations of succinate dehydrogenase (SDHx) are found in paragangliomas/pheochromocytomas [106]. Paragangliomas with SDHx mutations promote succinate accumulation, thereby inhibiting the α-ketoglutarate-dependent histone and DNA demethylases. This, in turn, establishes a hypermethylation phenotype, resulting in the downregulation of key genes involved in neuroendocrine differentiation and enhanced migration [163]. Another mechanism that mediates progression of tumors with FH and SDHx mutations relies on the intracellular accumulation of both fumarate and succinate, which consequently promote HIF1a stabilization [164,165] and execution of its transcriptional program.

Isocitrate dehydrogenases (IDHx) convert isocitrate to α-ketoglutarate. Among its three isoforms, mutations in the IDH1 and IDH2 genes are found in the majority of low- and high-grade gliomas [94] as well as in 20% of adults with acute myeloid leukemia (AML), and 5% of adults with myelodysplastic syndromes (MDSs) [166]. Instead of α-ketoglutarate, mutated IDH1 and IDH2 produce an oncometabolite, 2-hydroxyglutarate [103], which inhibits histone demethylases, causing the hypermethylation phenotype and hence altering the gene expression program [102]. IDHx mutations were shown to induce EMT in different tumor models by repressing the expression of miR-200 and thereby re-activating Zeb1, one of the key EMT transcription factors [167,168,169].

As glutamine is essential for cancer cells, it is not surprising that it has been shown to play an important role in EMT. Despite the fact that both enzymes, Glutaminases 1 and 2 (GLS1 and GLS2), are able to assimilate glutamine (convert it to glutamate and ammonia), apparently their levels and functions upon EMT are regulated differently. GLS1 was shown to be induced by TGF-beta and is essential for EMT in lung cancer cells, hepatocellular carcinoma [100], and breast cancer [101]. At the same time, expression of GLS2 was [170] inversely associated with EMT and decreased cancer stem-cell properties [171].

Lipid metabolism is another important aspect of EMT initiation. A number of studies have shown that FASN, a critical enzyme in fatty acid synthesis, induces EMT and cell invasion in various malignancies [172]. Noteworthy, TGF-beta was shown to also affect lipid metabolism [173].

Furthermore, beta-oxidation of fatty acids is an important process for EMT [174,175]. The corresponding enzyme, carnitine paltransferase 1A, promotes EMT in gastric [176], colorectal [177], and breast cancers [178].

Taken together, these data suggest complex relations between EMT and metabolic alterations in cancer (Figure 3). In the recent years, it has become apparent that EMT is tightly associated with metabolic reprogramming and itself can be induced by specific changes in metabolism and nutrients levels. In a poor nutrition environment, the competition between cancer cells for resources may result in generating phenotypic heterogeneity and selecting specific genetic and phenotypic subpopulations [155]. These changes would subsequently manifest in metabolic alterations associated with EMT.

## 6. Targeting Cancer-Associated Metabolic Alterations

Metabolic reprogramming taking part in malignant cells opens a window for therapeutic liabilities. A number of inhibitors of metabolic enzymes and carriers have been discovered (see comprehensive reviews on topic [8,117,179]). The nonexhaustive list of the inhibitors relevant to clinical applications is shown in Table 2 and is also displayed in Figure 4.

As rapidly proliferating cancer cells require high levels of DNA synthesis, one-carbon metabolism, which provides “building blocks” for this process, is the primary focus of therapy interventions.

In this respect, one-carbon metabolism has been a well-known therapeutic target for decades since Sydney Farber discovered amynopterin for the treatment of children with acute lymphoblastic leukemia (ALL). Later, amynopterin was replaced by methotrexate (MTX) which is a less toxic and more efficient pharmacological agent [61]. Up to now, MTX has been used as part of combined chemotherapeutic schemes to treat various malignancies. At the molecular level, MTX targets DHFR, thereby inhibiting the folate cycle. The latter consequently decreases purine and glutathione syntheses, as well as DNA and protein methylation. Amynopterin and MTX laid the groundwork for a new class of anticancer therapeutics called “antimetabolites”. Antimetabolites are small compounds which mimic folates or nucleotide precursors. By doing this, they inhibit enzymes that use the corresponding folates or nucleotide precursors as substrates. They are widely used to treat neoplasms of various origins [7].

For example, thiopurines (Thioguanine (6-TG), 6-Mercaptopurine (6-MP), and Azathioprine (Aza)) have been used to treat ALL and CML since the 1950s until the present time. In the human body, they are converted to structural analogs of guanine, which is a substrate for DNA polymerases. When incorporated into DNA during replication, it causes cytotoxicity mediated by the mismatch DNA repair (MMR) system [117].

Furthermore, gemcitabine, which inhibits ribonucleotide reductase (RNR), thus hampering the synthesis of deoxyribonucleotides required for DNA replication, is yet another clinically approved nucleoside analog. Another antimetabolite widely used in clinics is fluouracil (5-FU), a nucleoside analog of uracil [180]. 5-FU inhibits Thymidilate Synthase (TS), an enzyme that converts deoxyuridine monophosphate (dUMP) to deoxythymidine monophosphate (TMP).

Table 2 and Figure 4 summarize well-known inhibitors of one-carbon metabolism. The use of one-carbon metabolism inhibitors results in the inhibition of DNA and RNA syntheses, a decrease in gluthathione, S-adenosylmethionine, and certain amino acids, resulting in the limiting proliferation of malignant cells and induction of apoptosis. In addition to these well-known targetable enzymes, there are a number of other potentially important targets of one-carbon metabolism, e.g., serine hydroxymethyl transferase (SHMT2), methylene tetrahydrofolate reductase (MTHFD2), and inosine-5′-monophosphate dehydrogenase (IMPDH2), for which selective inhibitors are being discovered [7].

In addition to the one-carbon metabolism, lipid metabolism is also important and represents a potentially very promising target for cancer therapy. A number of studies have shown that FASN inhibition decreases proliferation of malignant cells, induces apoptosis, and prevents EMT in various cancer models [48]. Notwithstanding the fact that several inhibitors of FASN, including C75, cerulenin, and orlistat, have shown high efficacies in selective targeting of cancer cells, they have not been translated into clinics because of the severe side effects and toxicity they have caused to normal tissues [51]. However, the second generation of FASN inhibitors, TVB-2640, has recently entered clinical studies [181]. This provides certain hope that an anti-EMT drug will eventually be developed.

Regarding fatty acid beta-oxidation, the promising target is carnitine palmitoyltransferase-1 (CPT-1), a carrier protein that mediates the transport of fatty acids from the cytosol into mitochondria where the oxidation of fatty acids takes place [182]. One such inhibitor, Etomoxir, which went to the clinical trials as a treatment against diabetes and heart failure, displayed significant antineoplastic properties associated with the inhibition of fatty acid beta-oxidation [182,183,184].

Glycolysis is a very attractive pathway for anticancer therapy. 2-Deoxyglucose (2-DG), an inhibitor of the hexokinase 2 (HK2) enzyme, has demonstrated anticancer effects in different neoplastic models. However, it was not effective in small concentrations in clinical trials, whereas high doses which inhibited cancer were toxic [117]. Another well-studied inhibitor of glycolysis is 3-brompyruvate (3-BP). It targets several glycolytic enzymes including HK2, glyceraldehyde-3-phosphate dehydrogenase (GAPDH), and phosphoglycerate kinase (PGK-1). Although it was very effective in preclinical studies, 3-BP has demonstrated severe adverse effects even in animal models, limiting its participation to only several clinical trials [185,186]. One possible explanation for this is that the robust inhibition of glycolysis required for hampering of cancer growth may not be well-tolerated due to similar effects on the normal tissues that also rely on glucose metabolism [117]. There are number of other attempts to inhibit glycolysis by recently discovered inhibitors of several glycolytic enzymes [3]. However, none of them have entered clinical trials yet. 

While the possibility of targeting glycolysis as an anticancer therapy remains questionable, several studies describe clinical trials based on the inhibition of MCT-1 lactate carrier by a small molecule, AZD3956 [187,188]. MCT-1 belongs to the transmembrane monocarboxylate transporters (MCTs) family which mediates bidirectional transport of lactate outside and inside the cells [189]. MCT-1 is responsible for removing from cells an excess of lactate, thus preventing intercellular acidification due to the high rate of glycolysis [190]. Thus, inhibition of MCT-1 seems to be a promising therapeutic strategy. In this respect, diclofenac (a nonsteroidal anti-inflammatory drug) can be repurposed as a pan-MCT inhibitor. In line with this notion, Ordway with coauthors have shown that this drug was able to inhibit MCT activity, and reduce the Warburg phenotype and viability of cancer cells both in in vitro and in vivo models [191]. 

A promising inhibitor of cancer energy metabolism is lonidamine. It was first recognized as an antispermatogenic drug. Lonidamine was shown to have several targets to reduce the energy metabolism of cancer cells. It inhibits glycolytic enzyme HK2, lactate carrier MCT-1 and mitochondrial pyruvate carrier (MPC) [192,193]. Thus, lonidamine inhibits both respiration and glycolysis, leading to a decrease in cellular ATP levels. Importantly, lonidamine efficiently inhibits the cancer cells growth both in vitro and in vivo in combination with other anticancer therapeutics, including temazolomide [194,195,196]. It has successfully passed several clinical trials and is now used for cancer treatment in several countries [3]. However, the widespread application of lonidamine is limiting because of hepatic and pancreatic toxicities.

Another drug targeting respiration is antidiabetic therapeutic metformin. It inhibits the respiratory chain complex I by limiting oxidative phosphorylation and ATP production, which results in AMPK activation [197]. Metformin both decreases the risk of cancer and has been used as a neoadjuvant in numerous clinical trials [198].

The enzymes of TCA cycle, IDH1 and IDH2, are frequently mutated in glioblastoma multiforme and AML, thereby providing the clinical possibility to target these cancers. Mutations in these enzymes make them oncogenes due to the production of oncometabolites which induce epigenetic changes. A number of both IDH1 and IDH2 inhibitors have been developed and are subjected to ongoing clinical trials [149,199]. Edasitinib (AG-221), which targets IDH2, and Ivosedinib (AG-120), which targets IDH1, are approved for therapy of relapsed or refractory AML [200,201].

Glutaminase (GLS), which is responsible for the conversion of glutamine to glutamate, is potentially an attractive drug target for anticancer therapy [202]. CB-839 is a selective and orally bioavailable inhibitor of GLS, which displays significant antineoplastic activity in both cancer cells and animal models [44,45]. CB-839 is the only GLS inhibitor which has entered clinical trials to date [203].

## 7. Conclusions

Seventy years of successful antimetabolite therapy in clinics has demonstrated that cancer-related metabolism can be effectively targeted. However, important challenges still exist. In general, metabolism is a universal process occurring in every cell. However, metabolic alterations found in neoplastic cells may also present in normal, highly proliferating cells (for example, colon and lung epithelial). Therefore, an important issue of dosage of metabolic drugs should be taken into account to avoid undesirable side effects and excessive toxicity. Nevertheless, the high proliferation rate and particularities of metabolism of cancer cells opens an opportunity to predominantly targeting the neoplastic cells. The same is true for cytotoxic therapy based on DNA damage which is crucial mainly for dividing cells.

Cancer cells rely on their metabolic plasticity to be able to proliferate, disseminate throughout the body, and combat the immune system and anticancer therapy. In this review, we highlighted the key metabolic reactions that can be exploited by pharmacological intervention. However, it is important to remember that cancer cells have developed sophisticated regulatory mechanisms to fine-tune their metabolisms to adapt to various aggressive conditions. For example, upon EMT, cancer cells switch from the highly proliferative mode to a slow proliferating mode to alter their cytoskeletons and gain invasive properties. Thus, the pharmacological suppression of primary proliferating cancers may not be efficient against the EMT cells. Thus, suppressing EMT seems to be a promising approach to prevent metastases. However, it should be noted that this process also displays a high degree of plasticity. Once neoplastic cells extravasate and subsequently invade a tissue, they undergo the reverse process, MET. Therefore, interference with EMT could promote further metastasis instead of curing them if not targeted specifically at the site of origin.

To conclude, better understanding the “hallmarks” of metabolism in metastasizing malignant cells and cells upon colonization of new niches may open a new window of opportunities for the treatment of progressive cancers and discovery of new effective therapeutic approaches.

## Figures and Tables

**Figure 1 cancers-13-00762-f001:**
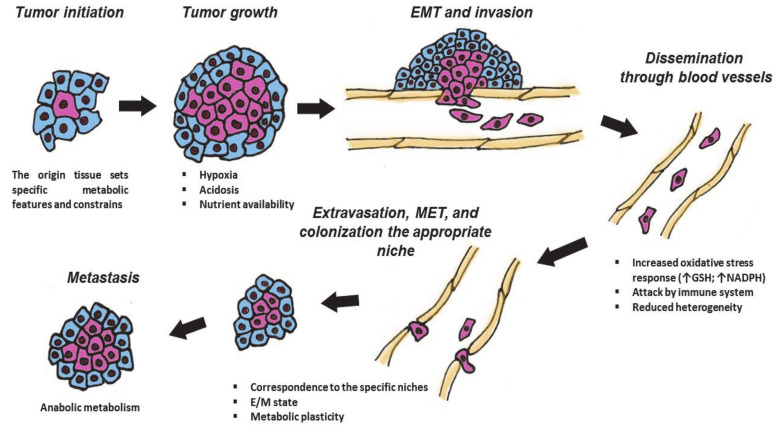
Different stages of tumor are associated with specific metabolic features. During the cancer development, neoplastic cells pass through several stages: tumor initiation, growth, intravasation into blood or lymph vessels, and dissemination across the body, extravasation, and Mesenchymal–Epithelial Transition (MET) to colonize appropriate niches. These events are associated with various challenges and require metabolic plasticity. GSH—glutathione; NADPH—Nicotinamide adenine dinucleotide phosphate; Arrows mean increasing the amount of GSH or NADPH; E/M—“hybrid” epithelial/mesenchymal state. Explanations are given in the text.

**Figure 2 cancers-13-00762-f002:**
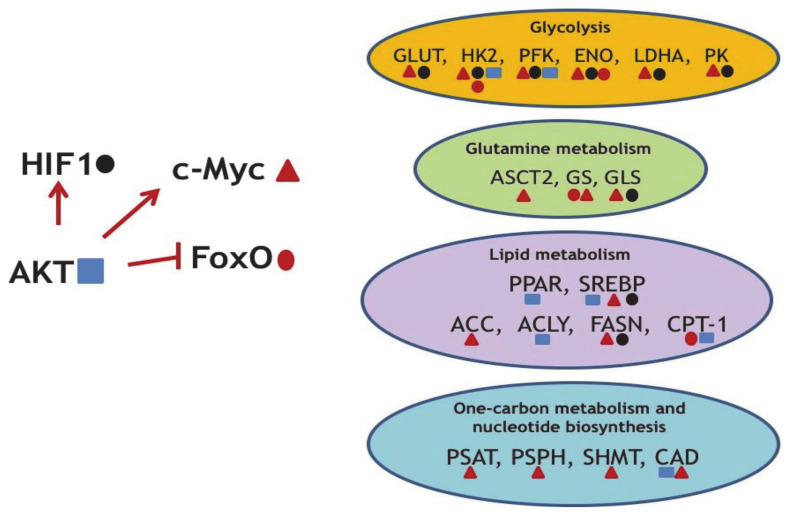
Several examples of the oncogene-mediated regulation of metabolic reprogramming. AKT upregulates oncogenic c-Myc and HIF1 and concomitantly downregulates the tumor suppressor FoxO. In turn, transcription factors HIF1, c-Myc, and FoxO (marked by specific geometric figures) regulate expression levels of enzymes indicated. ENO—enolase; PFK—phosphofructokinase; PK—pyruvate kinase; ASCT2—Solute Carrier Family 1 Member 5; PSAT—phosphoserine aminotransferase; PSPH—phosphoserine phosphatase; SHMT2—Serine Hydroxymethyltransferase 2; CAD—Carbamoyl-Phosphate Synthetase 2; Aspartate Transcarbamylase, And Dihydroorotase.

**Figure 3 cancers-13-00762-f003:**
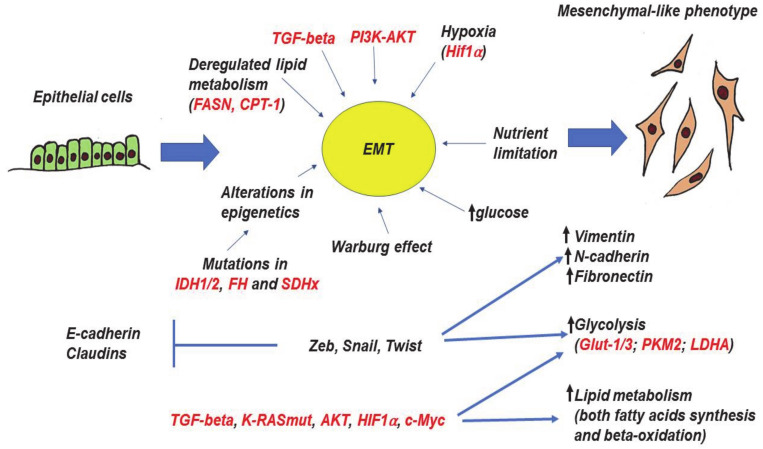
Epithelial-to-mesenchymal transition (EMT), its molecular drivers, and metabolic features. The process of EMT is triggered by various autocrine and paracrine factors, including TGF-β, PI3K, and AKT. In addition, certain metabolic factors (alterations of glucose level, deregulation of lipid metabolism, loss of function mutations in several TCA genes can also trigger EMT. The metabolic changes alter homeostasis of epigenetics through the accumulation of particular metabolites. EMT master-regulators (Zeb, Snail, and Twist families) as well as other inducers and coregulators of EMT (TGF-β, AKT, K-RASmut, HIF1α, c-Myc) are able to directly regulate glycolysis and lipid metabolism. All arrows mean “increase”, whereas “bars” mean inhibition. Red color—names of proteins, black color—processes and description. More explanations are given in the text.

**Figure 4 cancers-13-00762-f004:**
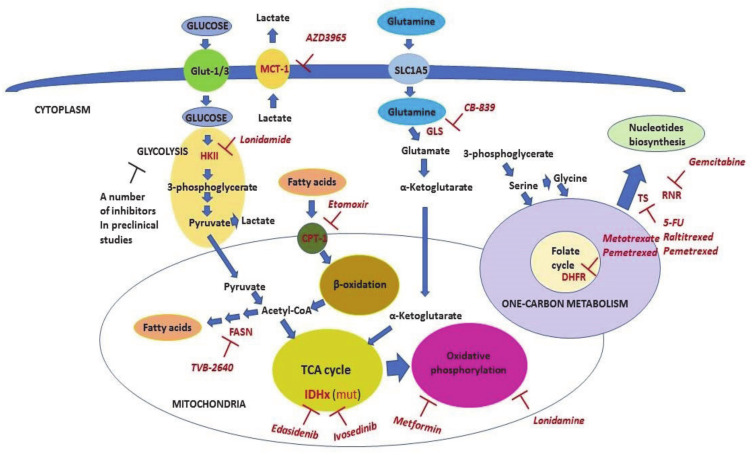
The scheme of metabolic pathways which are targeted in clinical oncology. The scheme shows metabolic pathways and the key enzymes (shown in red) involved in these processes. Inhibitors that are used as anticancer therapeutics (shown in pink) are also shown. Malignant cells consume energy mainly through utilizing glucose, fatty acids, and glutamine. The process of glycolysis begins with glycose phosphorylation and culminates with pyruvate, which then enters the TCA cycle in mitochondria. Pyruvate is also converted to lactate to maintain high levels of glycolysis. Lactate is exported from the cells mainly by Monocarboxylate carrier 1 (MCT-1) (which is inhibited by AZD3965 and lonidamide) to prevent intracellular acidosis. Fatty acids are transported inside mitochondria by Carnitine Palmitoyltransferase (CPT-1) (which is inhibited by Etomoxir). This step limits their beta-oxidation. The latter is the source of energy consumed by cancer cells via acetyl-CoA entering the TCA cycle. Additionally, cancer cells, unlike nonmalignant ones, are able to synthesize fatty acids. Fatty acid synthase (FASN) (which is inhibited by TVB-2640) mediates the limiting repeating step in this process. Mutated enzymes of the TCA cycle, isocitrate dehydrogenase 1 (IDH1) and IDH2, can be targeted by ivosedinib and edasidenib, respectively. Glutamine is converted to glutamate by Glutaminase (GLS) enzyme (which is inhibited by CB-839) and then to alpha-ketoglutarate in the next step reaction. Alpha-ketoglutarate, in turn, enters the TCA cycle and may be used for either energy production or anabolic metabolism (anaplerosis). The pivotal part of the one-carbon metabolism is the folate cycle, which can be inhibited by methotrexate and pemetrexed via inhibition of DHFR. The one-carbon metabolism provides one-carbon donor groups for biosynthesis of different compounds including nucleotides that are the rate-limiting factor for DNA synthesis and hence, proliferation of cancer cells. Fluouracil (5-FU), raltitrexed, and pemetrexed all inhibit Thymidylate Synthase (TS), whereas gemcitabine inhibits Ribonucleotide reductase (RNR). Detailed explanations are given in the text.

**Table 1 cancers-13-00762-t001:** Key metabolic regulators and their effects on some of their targets.

Metabolic Regulator	Pathway	Targets	Effect	Reference
**c-Myc**	Glycolysis	Monocarboxylate carriers (MCT1 and MCT2)	Upregulation	[71]
		Glucose transporter (GLUT1), hexokinase 2 (HK2) phosphoglucose isomerase (GPI), enolase (ENO), phosphofructokinase (PFK), glyceraldehyde-3-phosphate dehydrogenase (GAPDH), phosphoglycerate kinase (PGK), pyruvate kinase (PKM2), lactate dehydrogenase (LDHA)	Upregulation	[72,73]
	Glutamine metabolism	Glutamine transporters ASCT2 and SLC7A25	Upregulation	[73]
		Glutaminase (GLS)	Upregulation	[73,74]
	Lipid metabolism	ATP citrate lyase (ACLY), acetyl-CoA carboxylase (ACC), fatty acid synthase (FASN), and stearoyl-CoA desaturase (SCD)	Upregulation	[75,76]
	One-carbon metabolism	phosphoserine aminotransferase 1 (PSAT), phosphoserine phosphatase (PSPH), phosphoglycerate dehydrogenase (PHGDH), serine hydroxymethyl transferase (SHMT2)	Upregulation	[75,77]
**HIF1**	Glycolysis	Glucose transporters GLUT-1 and GLUT-3	Upregulation	[78]
		Hexokinase (HK2) phosphoglucose isomerase (GPI), enolase (ENO), glyceraldehyde-3-phosphate dehydrogenase (GAPDH), pyruvate dehydrogenase kinase 1 (PDK1)		[79,80]
	Fatty acids biosynthesis	Sterol regulatory element-binding protein 1 (SREBP-1), Fatty acid synthase (FASN), and Fatty acid-binding protein (FABP3,4 and 7)	Upregulation	[81,82]
	FAO	Peroxisome proliferator-activated receptor (PPARgamma)	Downregulation	[83]
		Glutaminase (GLS1)	Upregulation	[84]
**AKT**	Glycolysis	Hexokinase (HK2), pyruvate kinase (PKM2), phosphofructokinase (PFK)	Upregulation	[70,85,86]
	Fatty acids biosynthesis	Sterol regulatory element-binding protein 1 (SREBP-1), ATP-citrate synthase (ACLY)	Upregulation	[70,87,88]
	FAO	Carnitine palmitoyltransferase (CPT-1A)	Downregulation	[89]
**FOXO**	Fatty acids biosynthesis	Sterol regulatory element-binding protein 1 (SREBP-1)	Downregulation	[90]
	Lipolysis	adipose triacylglycerol lipase (ATGL)	Upregulation	[91]
	Glutamine metabolism	glutamine synthetase (GS)	Upregulation	[92]

FAO—fatty acid β-oxidation.

**Table 2 cancers-13-00762-t002:** Metabolic inhibitors, which are currently being used to treat cancer or participated in clinical trials.

Therapeutic	Target	Used to Treat Types of Cancer
Methotrexate	Dihydrofolate reductase (DHFR)	Breast cancer, leukemia, lymphoma, lung cancer, and osteosarcoma
Pemetrexed (Alimta)	Dihydrofolate reductase (DHFR), Thymidylate Synthase (TS)	Pleural mesothelioma and non-small cell lung cancer (NSCLC)
Raltitrexed (Tomudex)	Thymidylate Synthase (TS)	Malignant mesothelioma
Pralatrexate (Folotyn)	Dihydrofolate reductase (DHFR)	Relapsed or refractory peripheral T-cell lymphoma (PTCL) and aggressive type of non-Hodgkins lymphoma
5-FU (Adrucil)	Thymidylate Synthase (TS)	Colon, esophageal, stomach, pancreatic, breast and cervical cancer
Gemcitabine	Ribonucleotide reductase (RNR)	Breast, ovarian, nonsmall cell lung cancer, pancreatic and bladder cancer
AZD3965	Lactate carrier MCT-1	Diffuse large B cell lymphoma and Burkitt’s Lymphoma
Lonidamine	Hexokinase II (HKII), Lactate carrier MCT-1 Mitochondrial pyruvate carrier MPC	Brain tumors
Metformin	Complex I of respiratory chain	Breast, colon, pancreatic cancer, Chronic lymphocytic leukemia (CLL)
Etomoxir	Carnitine palmitoyltransferase-1 (CPT-1)	Type 2 diabetes and heart failure. Pre-clinical studies in cancer
TVB-2640	Fatty acid synthase (FASN)	Nonsmall cell lung cancer (NSCLC)
CB-839 (Telaglenastat)	Glutaminase (GLS)	Advanced renal cell carcinoma, Nonsmall cell lung cancer (NSCLC)
AG-221 (Edasidenib)	Isocitrate dehydrogenase (IDH2)	Acute myelogenous leukemia (AML)
Ivosedinib	Isocitrate dehydrogenase (IDH1)	Acute myelogenous leukemia (AML)

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
