# Peer review of "Linking Metabolic Reprogramming, Plasticity and Tumor Progression"

_cancers, 2021, doi:10.3390/cancers13040762_

Round 1
Reviewer 1 Report
I have carefully revised the manuscript entitled “Linking Metabolic Reprogramming, Plasticity and Tumour Progression” by Shuvalov et al. This review is focussed on a very hot topic regarding to the implication of the metabolic reprogramming in malignant cells and the potential use of anti-metabolic drugs as anti-cancer therapy. However, the review is not well presented. In general, very general explanations are reported, without a profound mechanistic insights regarding to the link between metabolic reprogram and plasticity and the impact on tumour progression. Figures 1 and 2 are not informative and in many cases EMT is wrongly mentioned for tumours that do not arise from epithelial cells.
Author Response
I have carefully revised the manuscript entitled “Linking Metabolic Reprogramming, Plasticity and Tumour Progression” by Shuvalov et al. This review is focussed on a very hot topic regarding to the implication of the metabolic reprogramming in malignant cells and the potential use of anti-metabolic drugs as anti-cancer therapy.
Our response: We appreciate the reviewer for acknowledgement of the topicality of our review manuscript.
In general, very general explanations are reported, without a profound mechanistic insights regarding to the link between metabolic reprogram and plasticity and the impact on tumour progression.
Our response: We have added more information on the molecular mechanisms regarding the oncogene-mediated impact on metabolic reprogramming (lines 240-322) and also prepared an additional figure (Figure 2) on this topic. Besides, we have expanded discussions throughout the manuscript by adding the pertinent information about the known molecular mechanisms (marked by yellow). These explanations are related to glycolysis and OXPHOS, lipid metabolic reprogramming, glutamine-mediated reprogramming as well as the regulation of metabolic reprogramming and its impact on cancer cells plasticity.
Figures 1 and 2 are not informative and in many cases EMT is wrongly mentioned for tumours that do not arise from epithelial cells.
Our response: We have prepared a new figure (Figure 2) illustrating the oncogene-mediated impact on metabolic reprogramming. We agree with the reviewer that EMT in cancer cells is not the right definition, because tumor cells display a high degree of plasticity and can express both epithelial and mesenchymal markers at the same time.
Reviewer 2 Report
I think this is well-written and organized review, focusing not only on the unique metabolic features of cancer cells, but also the plasticity of how cancer cells can switch their metabolic pathways during on their ‘journeys’ to different environments. The authors are able to convey a relatively complex picture to the readers in a ‘digestible’ manner. I am particularly appreciative of figure 3, which provides a concise summary of the key metabolic pathways discussed in this review, and how various pharmacological agents work.
I do not have any substantial suggestions for improvements other than pointing out the fact that there are some minor grammatical errors here and there.
Author Response
I think this is well-written and organized review, focusing not only on the unique metabolic features of cancer cells, but also the plasticity of how cancer cells can switch their metabolic pathways during on their ‘journeys’ to different environments. The authors are able to convey a relatively complex picture to the readers in a ‘digestible’ manner. I am particularly appreciative of figure 3, which provides a concise summary of the key metabolic pathways discussed in this review, and how various pharmacological agents work. I do not have any substantial suggestions for improvements other than pointing out the fact that there are some minor grammatical errors here and there.
Our response: We are grateful to the reviewer for careful reading of the manuscript.
Reviewer 3 Report
1- In general, it is a good review paper discussing the role of metabolic reprogramming and plasticity in cancer.
2-Introduction: the first paragraph is irrelevant and should be replaced by something related to metabolism and plasticity.
3- Line 64-65 doesn’t read well: “The structure of tumor is heterogenic because of different availability of and exposure to nutrients, oxygen, and aggressive immune cells constantly surveilling to combat tumor cells”
4- in above line please includes acidity that is a biproduct of Warburg effect and add citation related to it such as: Damaghi et. Al. PNAS 2021.
5- Line 75 is not clear: Not surprisingly, since metabolism provides energy and structural resources (“building blocks” for proliferation, glutathione and NADH for redox homeostasis, S-adenosyl methionine (SAM) – for epigenetic regulation, etc.) for all these functions of cancer cells, it is also very dynamically regulated.
6- In figure 1 there is no metabolic plasticity mentioned. You may add adaptation to hypoxia and acidosis at early stage and also switching to Warburg effect in later stage. Also, nutrient availability can define the metabolic reprogramming.
7- Remove the before more in line 85: …. is signified by the more than
8-Please add some examples of onco-associated regulating metabolic reprogramming in line 87-88. Onco-associate metabolic reprogramming such as?
8- Line 108 the references 14-15 are outdated and are not major publication in the field. Please use more up to date references.
9- line 115: please refer the hybrid EMT phenotype under hypoxia and acidosis.
10- Define ROS in line 127 since it is the first time it is mentioned.
11- Line 140: what does ‘Their’ refer to?
12- Line 140-145 need to be expanded in discussed in more details with proper references.
13- Line 146-156 needs to be discussed in more details and also mention some reprogramming switches involved in glutamine metabolism.
14- line 160-170: how lipid metabolism is regulated in metabolic reprograming and how it is related to plasticity of cancer cells needs to be discussed.
15-Line 196: of or mutation?
16- Line 196-198: the role of oncogenes in metabolic phenotype regulation needs to be expanded with examples and references.
17- line 216-219: the role of hypoxia, acidosis and Warburg effect in cancer initiation needs to be expanded with references.
18- Line 290 needs reference.
19- Line 291: acidosis doesn’t fully transform EMT and it induces partial EMT as shown in (Sadeghi et al frontiers in oncology 2020). This can add more level of plasticity to cancer cells. Please discuss this in the paragraph.
20- Line 327: the sentence needs to be corrected grammatically.
21- Line 338: the sentence needs to be corrected grammatically.
22- In figure 2 the role of acidosis and Warburg effect on EMT needs to be added.
23- Line 353: the two paragraphs about kRAS and myc don’t belong in this section. They should be moved up to oncogenes driven metabolitc regulation part earlier in the manuscript.
24- Line 458: please discuss the use of Diclofenac as pan inhibitor of MCT (ref Ordway et. Al. 2020)
Author Response
Suggestions for Authors
1- In general, it is a good review paper discussing the role of metabolic reprogramming and plasticity in cancer.
Our response: We appreciate the reviewer for the positive evaluation of our manuscript.
2-Introduction: the first paragraph is irrelevant and should be replaced by something related to metabolism and plasticity.
Our response: Done.
3- Line 64-65 doesn’t read well: “The structure of tumor is heterogenic because of different availability of and exposure to nutrients, oxygen, and aggressive immune cells constantly surveilling to combat tumor cells”
Our response: We have corrected this (lines 59-62)
4- in above line please includes acidity that is a biproduct of Warburg effect and add citation related to it such as: Damaghi et. Al. PNAS 2021.
Our response: We have added this reference.
5- Line 75 is not clear: Not surprisingly, since metabolism provides energy and structural resources (“building blocks” for proliferation, glutathione and NADH for redox homeostasis, S-adenosyl methionine (SAM) – for epigenetic regulation, etc.) for all these functions of cancer cells, it is also very dynamically regulated.
Our response: We have corrected this (lines 73-76).
6- In figure 1 there is no metabolic plasticity mentioned. You may add adaptation to hypoxia and acidosis at early stage and also switching to Warburg effect in later stage. Also, nutrient availability can define the metabolic reprogramming.
Our response: We have done it.
7- Remove the before more in line 85: …. is signified by the more than
Our response: Thank you, we have done this (line 89)
8-Please add some examples of onco-associated regulating metabolic reprogramming in line 87-88. Onco-associate metabolic reprogramming such as?
Our response: We have added appropriate examples (lines 81-83).
8- Line 108 the references 14-15 are outdated and are not major publication in the field. Please use more up to date references.
Our response: We have added new references (lines 108-110)
9- line 115: please refer the hybrid EMT phenotype under hypoxia and acidosis.
Our response: We have done it (lines 113-115)
10- Define ROS in line 127 since it is the first time it is mentioned.
Our response: We have done this (line 127)
11- Line 140: what does ‘Their’ refer to?
Our response: we have corrected this sentence (line 148).
12- Line 140-145 need to be expanded in discussed in more details with proper references.
Our response: We have done this (lines 139-171)
13- Line 146-156 needs to be discussed in more details and also mention some reprogramming switches involved in glutamine metabolism.
Our response: We have done this (lines 172-192)
14- line 160-170: how lipid metabolism is regulated in metabolic reprograming and how it is related to plasticity of cancer cells needs to be discussed.
Our response: We have done this (lines 220-233)
15-Line 196: of or mutation?
Our response: This sentence is excluded from the present version of the manuscript.
16- Line 196-198: the role of oncogenes in metabolic phenotype regulation needs to be expanded with examples and references.
Our response: We have done this (lines 252-302) and added Figure #2 and Table 1
17- line 216-219: the role of hypoxia, acidosis and Warburg effect in cancer initiation needs to be expanded with references.
Our response: We expanded this discussion (352-361).
18- Line 290 needs reference.
Our response: We expanded this section and added references (429-430).
19- Line 291: acidosis doesn’t fully transform EMT and it induces partial EMT as shown in (Sadeghi et al frontiers in oncology 2020). This can add more level of plasticity to cancer cells. Please discuss this in the paragraph.
Our response: We have briefly discussed this (lines 432-434)
20- Line 327: the sentence needs to be corrected grammatically.
Our response: We have corrected it (lines 470-471).
21- Line 338: the sentence needs to be corrected grammatically (now line 480).
Our response: We have done it (line 480).
22- In figure 2 the role of acidosis and Warburg effect on EMT needs to be added.
Our response: We have done it.
23- Line 353: the two paragraphs about kRAS and myc don’t belong in this section. They should be moved up to oncogenes driven metabolitc regulation part earlier in the manuscript.
Our response: We have moved them to the section “2. Specific metabolic alterations in malignant cells”
24- Line 458: please discuss the use of Diclofenac as pan inhibitor of MCT (ref Ordway et. Al. 2020)
Our response: We have discussed it (Line 586-589)
Round 2
Reviewer 1 Report
I have carefully revised the manuscript entitled “Linking Metabolic Reprogramming, Plasticity and Tumour Progression” by Shuvalov et al. The review has included significant information on the molecular insights and a new figure has been included to clarify some aspects of the manuscript.
Reviewer 3 Report
The manuscript has been improved significantly.